# Heat Stress in Cotton: A Review on Predicted and Unpredicted Growth-Yield Anomalies and Mitigating Breeding Strategies

**Sajid Majeed [1], Iqrar Ahmad Rana [2], Muhammad Salman Mubarik [2], Rana Muhammad Atif [1], Seung-Hwan Yang [3], Gyuhwa Chung [3,*], Yinhua Jia [4], Xiongming Du [4], Lori Hinze [5] and Muhammad Tehseen Azhar [6,7,*]**

[1]    Department of Plant Breeding and Genetics, University of Agriculture, Faisalabad 38400, Pakistan; sajidmajeedpbg@gmail.com (S.M.); dratif@uaf.edu.pk (R.M.A.)
[2]    Centre of Agricultural Biochemistry and Biotechnology, University of Agriculture, Faisalabad 38400, Pakistan; iqrar_rana@uaf.edu.pk (I.A.R.); msmubarik@gmail.com (M.S.M.)
[3]    Department of Biotechnology, Chonnam National University, Chonnam 59626, Korea; ymichigan@jnu.ac.kr
[4]    State Key Laboratory of Cotton Biology, Institute of Cotton Research Chinese Academy of Agricultural Science, Anyang 455000, China; jiayinhua_0@sina.com (Y.J.); duxiongming@caas.cn (X.D.)
[5]    US Department of Agriculture, Agricultural Research Service, College Station, TX 77845, USA; lori.hinze@usda.gov
[6]    School of Agriculture Sciences, Zhengzhou University, Zhengzhou 450000, China
[7]    Institute of Molecular Biology and Biotechnology, Bahauddin Zakariya University, Multan 60800, Pakistan
*    Correspondence: chung@chonnam.ac.kr (G.C.); tehseenazhar@gmail.com (M.T.A.)

**Abstract:** The demand for cotton fibres is increasing due to growing global population while its production is facing challenges from an unpredictable rise in temperature owing to rapidly changing climatic conditions. High temperature stress is a major stumbling block relative to agricultural production around the world. Therefore, the development of thermo-stable cotton cultivars is gaining popularity. Understanding the effects of heat stress on various stages of plant growth and development and its tolerance mechanism is a prerequisite for initiating cotton breeding programs to sustain lint yield without compromising its quality under high temperature stress conditions. Thus, cotton breeders should consider all possible options, such as developing superior cultivars through traditional breeding, utilizing molecular markers and transgenic technologies, or using genome editing techniques to obtain desired features. Therefore, this review article discusses the likely effects of heat stress on cotton plants, tolerance mechanisms, and possible breeding strategies.

**Keywords:** breeding; climate change; genetics; heat stress; upland cotton

## 1. Introduction

Upland cotton (*Gossypium hirsutum*) is a multipurpose cash crop. The lint of cotton is the major product of this crop and is used by the textile industry for cloth manufacturing. Cotton is cultivated on an area of about 34.1 million hectares with a production of 126.5 million bales, and it is grown in more than 35 countries [1]. India is the largest grower and producer of cotton, with production of ~6.1 million tons of cotton every year. Other leading cotton producing countries include China, United States, Brazil, Pakistan, Turkey, and Uzbekistan. China is the largest consumer of cotton, with consumption of ~7.60 million tons of cotton annually [2]

From sowing to harvesting, the cotton crop faces numerous problems including infestation of insect pests, diseases, heat, drought, cold and salinity stresses, trash during picking, and post-harvest management problems [3–6]. Each of these causes significant reduction in yield and quality of cotton fibres. Therefore, comprehensive research on each aspect is required in order to understand these problems. The present discussion focuses on high temperature stress and on minimizing the losses due to this abiotic stress. Thus, a

detailed review about the effects of heat stress on cotton plants and possible strategies for its mitigation is described in the following paragraphs.

Heat stress is often called high temperature stress. It is one of the limiting factors in crop productivity. Heat stress is defined as a condition when the temperature is high enough for a sufficient period of time to cause irreversible damage to plant development and functions [7]. A sudden increase of 5–7 °C in maximum temperature for a few days with a corresponding increase in ambient minimum temperature causes "high temperature stress" in plants. The temperature requirement varies from species to species, and it also depends upon time of exposure, intensity of exposure, air or soil temperature, night/day temperature, and age of the plant. Therefore, a particular temperature cannot be defined as a cardinal point for heat stress. Generally, cool season plant species are more vulnerable to heat stress than compared to warm season crops [7,8]. Moreover, plants of similar species adapted in different climatic regions have different degrees of temperature tolerance. For example, cotton grown in the United States and China is considered to be under heat stress when temperatures increase above 38 °C, while in Pakistan and India this temperature is considered optimum and temperatures greater than 46 °C are considered as heat [9–11]. Cotton responses to heat stress are presented in detail below.

Traditional breeding strategies have been utilized to incorporate heat stress tolerance into upland cotton. Most of the improved cultivars and breeding lines are the outcome of purely classical breeding as very little molecular and genomic tools have been used to date. Such breeding efforts were based on intensive selection, which reduced genetic variability in cotton. Mutagenic agents have been used to create variability and have resulted in the release of high-yielding cultivars, including NIAB-78 as a successful example from Pakistan [12,13]. Due to the increased resources needed to screen large populations, reduced frequency of desirable alleles, and pleiotropic effects, mutation breeding has gradually been replaced with marker assisted breeding and site-specific mutagenesis technologies. The use of advanced genomics and biotechnological tools has also become important as the challenges to cotton production escalating. Later in this review, advanced breeding tools are discussed that can be utilized to mitigate the effect of changing climatic conditions, especially high temperature stress.

## 2. Effects of High Temperature Stress on Cotton

High temperatures in arid and semi-arid regions of the world are inducing negative impacts on growth, development, and productivity of several field crops [14]. Heat stress can cause damage to a cotton plant in almost every stage of its life, but it is reported that the reproductive stages of cotton are more sensitive to high temperature than compared to vegetative growth stages [15]. Both day and night temperatures play an important role in determining yield potential in crop plants, but high night temperature reduces yield and causes significant damage, while the role of high day temperature is secondary [16]. The adverse effects of high night/day temperature on different plant stages are shown in Figure 1.

### 2.1. Effects on Germination of Cotton Seed

Successful germination of seed and development of seedlings requires a good soil environment, especially soil moisture content and soil temperature. These requirements vary from plant species to species [17,18]. Generally, high temperature results in poor seed germination in field crops. The germination of seed is a complex physiological process that depends upon the activity of several cellular organelles and enzymes. These enzymes need to be produced in a continuous manner to perform various activities related to metabolic processes. For instance, in maize at high temperatures, these enzymes denature or slow down these activities, which results in reduced metabolism processes [19]. Moreover, high soil temperature also increases the rate of transpiration, which reduces the water availability to seed. Less availability of water also slows down the seed germination [20]. The optimal temperature of 28 to 30 °C is needed for germination of cotton seed. As

the temperature decreases, the rate of seed germination also decreases, and very poor germination can be observed at temperatures <20 °C. Similarly, increases in temperature from the optimal range ($\geq$38 °C) have also been reported to result in decreased germination of cotton seed [21].

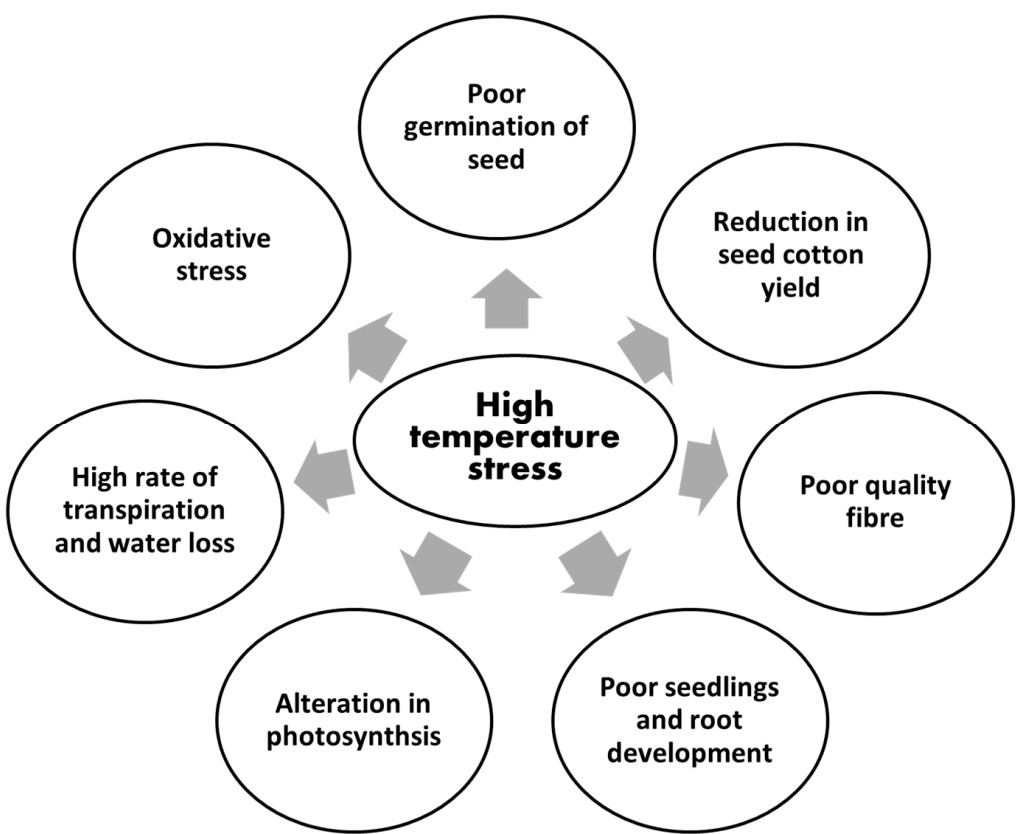

**Figure 1.** Salient adverse effects of high temperature stress on cotton plants.

*2.2. Effects on Early Vegetative Growth Stages*

In addition to germination, uniform and rapid emergence of seedlings from soil is necessary to achieve vigorous growth of plants and high yield [22]. High temperature stress during the early stages of plant growth affects the emergence of seedlings and results in the development of poor seedlings. In a previous experiment, researchers grew seedlings of cotton under various temperature regimes, i.e., 38 °C and 32 °C for 8 days. They recorded 152 mm shoot and 173 mm root growth at 32 °C temperature, while at 38 °C only 50 mm shoot and 86 mm root growths were observed [17]. The emergences and growths of cotton seedlings were also assessed for four different temperatures, i.e., 20, 30, 40, and 50 °C. The maximum emergence of seedling with vigorous growth was observed at 30 °C. The genotypes developed for hot tropical environments with higher seed weight showed vigorous seedlings even at 40 °C than compared to genotypes with smaller seed size and weight. Cotton seedlings did not emerge at 50 °C [23].

Along with the above ground parts of plants, roots are also severely affected by high temperature stress. The damaging of roots minimizes their uptake of nutrients and water from soil that can disturb the entire physiological mechanism of the plant and limit its productivity. The cotton plant has a taproot system. Studies showed that cotton roots develop poorly under high soil temperature and result in a poor crop stand [24]. A study was conducted on 10 upland cotton genotypes in order to determine the heat stress ability of roots by measuring related parameters. After 120 min of seed germinations, the seedlings were subjected to seven different temperature regimes between 25 to 45 °C. The results revealed that all the genotypes showed normal root growth up to 35 °C. The roots' growth

declined when temperature exceeded from 35 °C, and irreversible damage to roots of all the cultivars occurred at 45 °C [25].

### 2.3. Effects on Lateral Vegetative Growth Stages

Upland cotton is generally sown at the start of the summer season in tropical and subtropical parts of the world. Therefore, seedlings of cotton experience lower temperatures than compared to later lateral growth stages. The cotton crop normally experiences its highest temperature at bud initiation. This is the first flowering stage that causes instability in production [26,27]. A significant reduction in the number of fruiting branches has been observed under heat stress [28]. Dropping of floral buds, flowers, and bolls are common when the temperature increases above average. The retention of fruits on fruiting branches declined rapidly at temperatures increased to ≥33 °C in Mississippi, USA [29]. The shedding of fruiting bodies is a natural mechanism of the cotton plant for decreasing the fruit load in order to adjust the supply and demand balance of nutrients and water in the plant. Generally, the cotton plant drops ~50% premature bolls, but this percentage increases with increases in temperature. Researchers claimed that retention of bolls until harvest is the bigger challenge than compared to increasing the total number of fruiting bodies or bolls per plant. They indicated that the best possible strategy to increase cotton production is to minimize flower and boll shedding [30].

Boll weight and boll size are also reduced under heat stress conditions. Temperatures greater than 40 °C result in shortening the boll maturation period and formation of smaller sized bolls having less weight than bolls developed under normal temperatures [31]. An experiment was carried out with 23 accessions of cotton, which were grown under normal and heat stress conditions for two years. The boll weight of all the genotypes was reduced under the heat stress condition [32]. The reduction in boll weight of cotton cultivars grown under high temperatures was also observed by other researchers [33,34]. It was found that foliar application of ascorbic acid increased the heat tolerant ability of plants and minimized losses due to heat stress. Zeiher and Brown conducted several experiments under various environments, including greenhouse, growth chamber, and field environments. They concluded that boll size, fruit retention percentage, and number of seeds per boll were also reduced when temperatures were above 30 °C [29,35–37].

### 2.4. Effects on Yield and Quality of Fibre

Fibre production upon maturity is the ultimate goal of growers. Both the quantity and quality of fibre determine the end value of the cotton crop. The yield of seed cotton is a complex trait, which is the result of various morphological and physiological features of the plant. It also has a strong correlation with antioxidant activities. Both of these parameters are polygenic in nature and highly influenced by environmental conditions [38]. The negative association of fibre quantity with its quality is another challenge for breeders in terms of improving both features simultaneously [39]. Numerous studies have reported the effect of high temperature on fibre yield and quality [40–43]. An experiment was conducted on three cotton cultivars grown in nine diverse environments in order to investigate the effects of heat stress. The results revealed that lint yield and quality attributes were severely affected by high temperature. It was further concluded that cotton plants can perform well within specific temperature ranges as low temperature also adversely affected the cotton yield. The minimum threshold temperature of 22 °C was recorded to maintain seed cotton yield [44]. Another experiment revealed that both short and long term increases in day temperature decrease the biomass of cotton fruits, which ultimately results in low yield and poor quality fibres [45]. An increase of 2–3 °C temperature from the optimal temperature (32 °C) in Nanjing, China, resulted in a decrease of 10% of biomass, while yield declined by 40%. The results also revealed that the micronaire value of fibre increased under heat stress, which results in coarse fibres with less strength [46]. The developmental process of cotton fibres is sensitive to increases in temperature. Such conditions also reduce the time required for boll maturation that results in short fibres. High temperature conditions

lasting up to 5 days did not change fibre quality, but the prevalence of these conditions for more than a week may cause irreversible damage to cellulose and significant reduction in fibre quality [47].

### 2.5. Effect on Floral Parts

Information about pollen viability and stigma receptivity is important for increasing productivity because effective pollination is essential for fruiting and seed setting in crop plants [48]. The germination and length of pollen tubes in cotton were monitored in 12 cultivars grown under different temperature regimes. The results indicated that maximum pollen germination and longest pollen tube length were observed at 32 °C. The pollen failed to germinate at 47 °C, while no pollen tubes formed above 44 °C [49]. Another experiment revealed that length of the pollen tube decreases as temperature increases above 32 °C, while germination of pollen decreases above 37 °C. Therefore, it can be suggested that the formation of the pollen tube is more sensitive to high temperature stress than compared to pollen germination. *G. barbadense* was found to be more sensitive to heat stress. A lower in vitro pollen germination percentage was recorded for heat treated pollen of pima cotton than compared to upland cotton [50]. The length of the filament was significantly reduced when cotton flowers were exposed to high temperature stress, and this resulted in the appearance of an elongated stigma. The actual length of stigma remained the same, but the length of filaments were reduced to the extent that the stigma appeared to be very long, and the process of self-pollination was badly affected [29]. The loss of receptivity of the stigma under high temperature is also reported in sweet cherry and peach [51,52]. The penetration of pollen grains to the ovule via the pollen tube ensures the successful fertilization process, and the penetration of pollen tube through the stigma, style, and ovule at high temperatures was reduced, which resulted in poor seed setting [53]. The effects of heat stress on various reproductive phases of plants are illustrated in Figure 2.

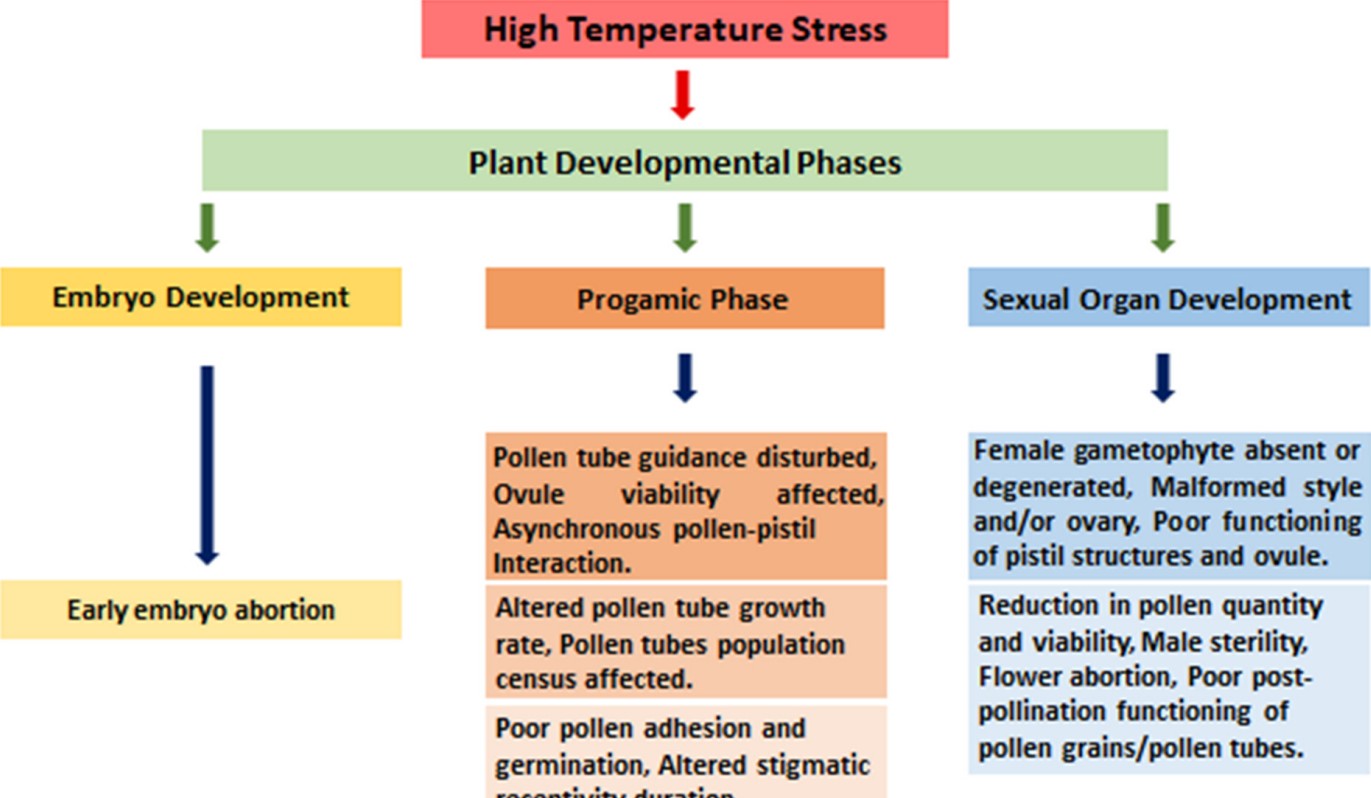

**Figure 2.** Effects of high temperature stress on reproductive phases of plant.

## 2.6. Effects on Physiological Attributes

Plant growth and development is based on physiological processes such as photosynthesis, membrane permeability, and stomatal conductance [54]. Heat stress adversely affects physiological attributes of plants and limits productivity [55]. The rate of photosynthesis is significantly reduced or even inhibited at high temperatures. The deactivation of ribulose 1,5 bisphosphate carboxylase/oxygenase (Rubisco) and increase in ionic conductance of thylakoid membranes are the primary causes of photosynthesis reduction or inhibition in cotton during high temperature conditions [56,57]. Chlorophyll content also decreases when cotton plants are exposed to high temperature that results in a decrease in the rate of photosynthesis [58]. Heat stress changes the permeability of membranes and alters cell differentiation and elongation by causing injuries to cellular membranes and deforming the organization of microtubules and cytoskeleton [59]. Higher cell membrane thermostability is positively associated with heat tolerance in cotton. Therefore, a number of experiments have been conducted to screen for heat tolerant accessions on the basis of cell membrane thermostability [60–62]. Stomatal conductance is directly related to water relations and photosynthesis in plants. High temperature causes opening of stomata and an increase in stomatal conductance that results in a decrease in the water potential of leaf. High stomatal conductance also increases the rate of transpiration and intercellular $CO_2$. Stomatal conductance increases up to 40% in most of plant species when temperature rises from 30 to 40 °C [63]. The advantage of higher stomatal conductance is associated with cooling of leaves, which provides tolerance to heat stress [64]. Experimental results have shown that upland cotton has more stomatal conductance and higher rate of photosynthesis under high temperature conditions than compared to pima cotton [65]. Studies revealed that differences in cotton accessions and species for stomatal conductance are under genetic control. Thus, this trait can be improved through breeding and selection [66–68].

## 3. Mechanisms of Heat Tolerance

### 3.1. Antioxidant Activity in Response to Oxidative Stress

Crop plants face environmental stresses which alter the various metabolic activities in order to ensure balance between production and consumption of energy through oxidation and reduction reactions [69]. This change in metabolism alters the concentration of various molecules. Likewise, metabolic imbalances during high temperature stress promote the extra-accumulation of reactive oxygen species (ROS) into the cellular compartment of plants [70]. ROSs are highly reactive chemicals formed from $O_2$. Excessive production and over-accumulation of ROS in plant cells cause irreversible damage to its organelles through oxidative stress. However, a balanced amount of ROS is required for normal activities such as detoxification of poisonous substances, antimicrobial phagocytosis, and apoptosis. ROS also benefits plants by acting as signaling molecules for activation of numerous genes related to stress tolerance, cell proliferation, seed germination, growth of root hairs, and cell senescence [71,72]. The over-accumulation of ROS during stress conditions results in the oxidation damage of vital molecules such as DNA, proteins, and lipids. This condition is termed as oxidative stress in plants [73]. ROS includes free radicals such as the hydroxyl radical (OH) and superoxide anion ($O_2^-$), as well as non-radicals such as singlet oxygen ($^1O_2$) and hydrogen peroxide ($H_2O_2$). These species are produced by excitation and reduction in intra-cellular oxygen ($O_2$). The schematic diagram of ROS production is illustrated in Figure 3.

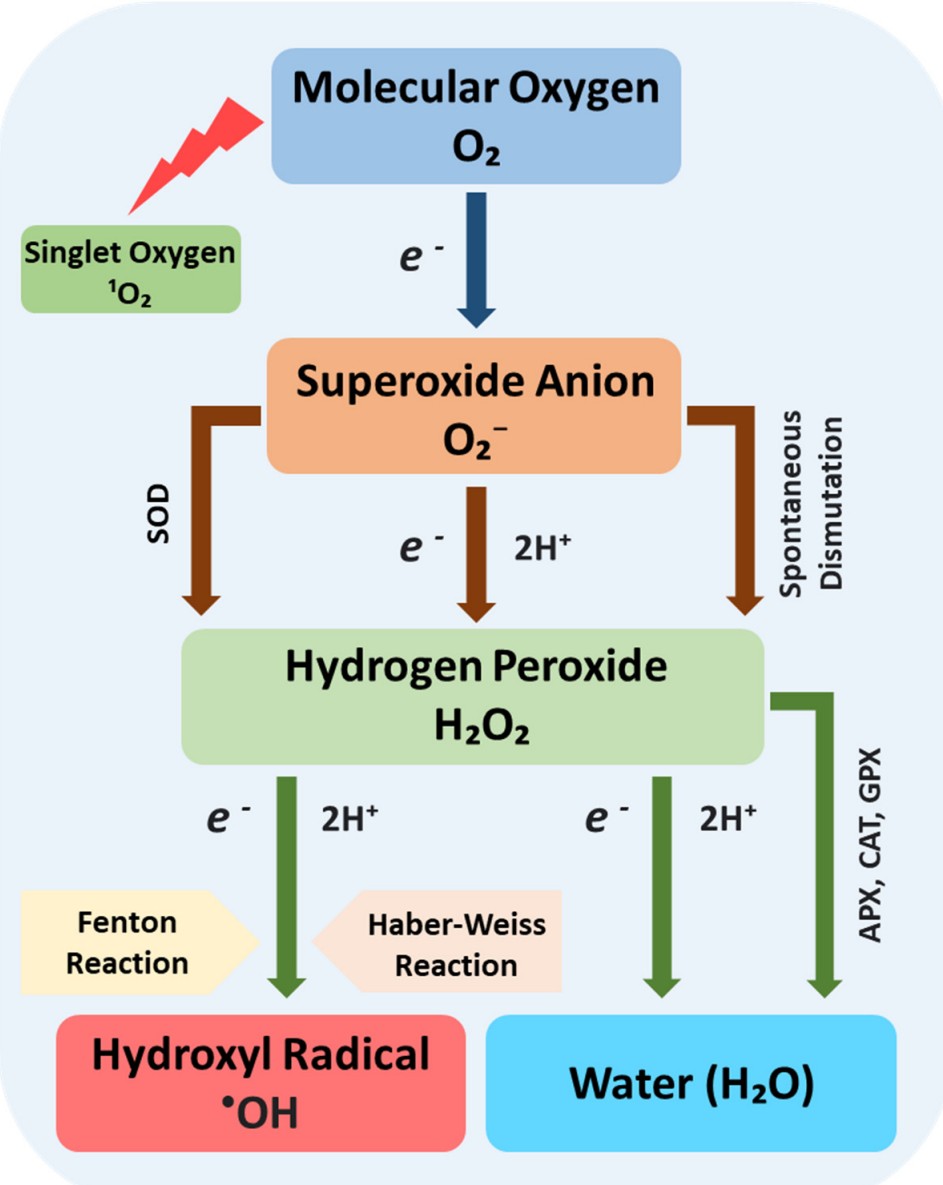

**Figure 3.** Schematic diagram of ROS production in plants.

The concentration of ROS and scavenging capacity of antioxidants in cotton is considered as a selection criterion for heat tolerant accessions [74]. An experiment was conducted using two cotton cultivars, and heat stress was applied gradually from 30 to 45 °C on 30 days old seedlings. The results revealed a 206 to 248% increase in hydrogen peroxide content and a 40 to 170% increase in lipid peroxidation under heat stress conditions. The concentration of non-enzymatic antioxidants also increases with increases in temperature, while the activity of enzymatic antioxidants such as superoxide dismutase (SOD), catalase (CAT), peroxidase (POX), and ascorbate peroxidase (APX) increase 56–70%, 37–69%, 43–91%, and 22–27%, respectively. It was concluded that genotypic differences exist in cultivars for ROS production and antioxidants response. Higher levels of antioxidants and lower levels of ROS during high temperature are an indication of heat stress tolerance [75]. In another study, the cotton plants were grown at two temperature regimes, i.e., 38 and 45 °C. The results indicated non-significant differences in the concentration of hydrogen peroxide at both temperatures, while the concentration of proline decreased rapidly and significantly as the temperature increased from 30 to 45 °C. The activity of

SOD declined at 45 °C while the activity of CAT, POX, and APX increased with the increase in temperature [76]. It is found that the exogenous application of hydrogen peroxide on cotton plants triggers the activity of SOD and CAT. It was further concluded that foliar applications of $H_2O_2$ on field grown cotton can enhance the heat tolerant ability without compromising yield [77]. The effect of high night temperature on biochemistry of leaf and pistil was studied in upland cotton cultivar. The results indicated that glutathione reductase activity in leaves is increased with an increase in night temperature, while no change in the concentration of glutathione reductase was observed in pistils. This shows that the antioxidant mechanism of pistil or floral parts is less sensitive to changes in mean night temperature than compared to leaves or vegetative parts of the cotton plant [78].

### 3.2. Role of Heat Shock Proteins

Heat shock proteins (HSPs) are important contributors to cellular homeostasis under heat stress. These molecular chaperons are upregulated during high temperature conditions and perform various activities in order to maintain the integrity of the cell [79]. On the basis of molecular weight, HSPs are divided into five major groups, i.e., small HSPs, HSP60, HSP70, HSP90, and HSP100 [80]. The details of each group are summarized in Table 1. Each HSP group is unique in nature and specific in function. Small HSPs (sHSPs) have low molecular weight (12–40 kDa) and are the most diverse in nature with respect to cellular location, function, and sequence similarities [81]. These sHSPs bind to non-native proteins, prevent non-native aggregation through hydrophobic interactions, and facilitate their refolding by ATP-dependent chaperons such as ClpB/DnaK [82]. Almost all sHSPs have an α-crystallin domain, which forms a dodecamer double ring and helps in the folding of proteins [83]. Previous work has revealed that the expression of the sHSP coding gene, i.e., *Hsp 17.7*, is directly related to thermal stress tolerance in plants [84]. A quantitative expression analysis of *GHSP26* (a small HSP coding gene in cotton) indicated that the leaves of cotton have 100-fold increased concentration of proteins encoded by this gene during water deficit conditions [85].

**Table 1.** Characterization of various groups of HSP in plants.

| Group (Sub-Families) | Representative Members | Intracellular Localization | Major Role | Reference |
|---|---|---|---|---|
| sHSP | | | | |
| I | Hsp17.6 | Cytosol | | |
| II | Hsp17.9 | Cytosol | Stabilization of non-native proteins and prevent aggregation | [86,87] |
| III | Hsp21, Hsp26.2 [5] | Chloroplast | | |
| IV | Hsp22 | Endoplasmic reticulum | | |
| V | Hsp23 [5] | Mitochondria | | |
| VI | Hsp22.3 | Membrane | | |
| HSP60 | | | | |
| Group I | Cpn60 [2] | Chloroplast, mitochondria | Folding of proteins | [88,89] |
| Group II | CCT [3] | Cytosol | | |
| HSP70 | | | Protein import, signal transduction, transcriptional activation, assist refolding and prevent aggregation of proteins | |
| DnaK | Hsp/Hsc70 | Cytosol | | |
| | Hsp70 | Chloroplast, mitochondria | | [90–93] |
| | Bip [1] | Endoplasmic reticulum | | |
| Hsp110/SSE | Hsp91 | Cytosol | | |
| HSP90 | AtHsp90-1 | Cytosol | Facilitate in genetic buffering and maturation of signaling molecules | [94–96] |
| | AtHsp90-5 | Chloroplast | | |
| | AtHsp90-6 | Mitochondria | | |
| | AtHsp90-7 | Endoplasmic reticulum | | |
| HSP100 | | | Unfolding and disaggregation of proteins | |
| Class I | ClpB, ClpA/C, ClpD | Cytosol, mitochondria | | [97,98] |
| Class II | ClpM, ClpN, ClpX, ClpY | Chloroplast | | |

HSP60 is generally known as a mitochondrial chaperon or chaperonin 60. It plays two essential roles in mitochondria during high temperature conditions, i.e., maintenance of the unfolded state of proteins for their transportation across the inner mitochondrial membrane and the folding of important proteins into a matrix [88]. HSP60 is also involved in assisting proteins that help in photosynthesis such as Rubisco [99]. Studies revealed that a mutation in *Chaperonin-60α* gene that codes for HSP60 protein causes a defection in chloroplasts, which ultimately results in poor seedlings and embryo development in *Arabidopsis* plant [100]. However, deletion of this gene results in cell death [101]. It has been experimentally verified that transgenic tobacco plants with reduced *Cpn60β* (chaperonin 60β) exhibited phenotypic defects such as delayed flowering, stunted growth, and leaf chlorosis [102]. HSP70s are considered important cellular machinery involved in the folding of proteins and in preventing their aggregation [103]. The overexpression of HSP70s is an indication of heat tolerance in plants. It is reported that *HSP70* genes of cotton play essential roles during fibre development. The inhibition of these genes results in the retardation of fibre elongation. The inhibition of *HSP70* genes results in oxidative stress by elevating the level of $H_2O_2$, which causes damage to epidermal layers of the ovule [104]. HSP70 proteins also act as signaling molecules for transcriptional activation and de-activation [91].

HSP90 proteins are quite distinct from other chaperons because most of them are substrates involved in signal transduction, such as signaling kinases and hormone receptors [105]. They also manage the folding of proteins [106]. HSP90s are among the most abundant proteins of the cell (1–2% of the total), are constitutive in nature, and act together with HSP70s as multi-chaperone machinery. The expression of HSP90 proteins increases significantly during hot conditions [107]. HSP100 belongs to the AAA ATPase family and performs various functions such as unfolding and disaggregation of proteins [108]. In addition to heat stress tolerance, HSP100 also performs housekeeping functions in the cell, including the development of chloroplasts [109,110].

### 3.3. Small RNAs Activity in Regulating Heat Stress

With the advancement in high throughput sequencing technologies, plant researchers have revealed various roles of microRNA (miRNA) and small interfering RNA (siRNA) during biotic and abiotic stress conditions. These small RNAs are actively involved in the degradation of mRNA and prevent translation of various proteins in plant cells [111–113]. The regulatory roles of various miRNAs have been characterized during high temperature conditions in many plant species, including chestnuts and *Arabidopsis*, e.g., the miR156 and miR157 families comprise 17 and four miRNAs, respectively. These miRNAs upregulate during hot temperature conditions and target *SPL* genes, which are essential for floral development in plants. Thus, flowering is controlled by these miRNAs when plants are exposed to high temperature conditions [114,115]. Over-production of miR157 targets *SPL* genes in cotton during heat stress results in a reduction in flowers and smaller sized bolls with fewer seeds [116]. The similar role of these miRNA families is reported in *Brassica rapa* [117], citrus [118], and *Arabidopsis* [119].

Experiments revealed that the expression of miR159 is down-regulated in heat tolerant genotypes of wheat upon exposure to hot temperature regimes. This miRNA acts as a negative regulator of *MYB* transcriptional factors [120]. The major role of the auxin response factors (*ARFs*) gene family is the regulation of auxin levels in plants. The over-expression of miR160 in cotton increases its susceptibility to high temperature stress by suppressing the expression of *ARF* genes [121]. It is found that the expression of miR162 increases up to 15-fold during drought conditions. Increases in concentration of miR162 under heat and salinity stresses is reported in cotton and rice [122,123]. This microRNA controls the transcription of numerous genes by targeting zinc finger proteins (ZFPs) and acts as a regulator of dicer such as proteins [124,125]. The expression level of miR164 was observed to decrease 0.3-fold under heat stress conditions. It targets *HSP17* genes and also regulates the expression of various genes essential for mitogen activated protein kinase (*MAPK*) mediated signaling pathways and the activation of *NAC* transcriptional factors in wheat,

rice, and alfalfa [126–128]. It is reported that expression of miR171 increased several folds upon exposure of the plant to high temperature. It targets *GRAS* genes, which are involved in various developmental processes, i.e., flowering time, floral meristem determinacy, plant height, and leaf architecture in cotton [129,130].

The regulation of flowering and floral organs is controlled by *AP2* genes [131]. This gene family is regulated by miR172, as reported in roses [132]. Thus, up-regulation and downregulation of these miRNAs is directly related to flowering timing and the transition of floral and vegetative phases in rice and *Arabidopsis* during high temperature conditions [133,134]. The miR390 of cotton controls the formation of lateral roots by targeting *ARF* genes [135]. The miR393 is considered a regulator of auxin receptors [136]. Overexpression causes a delay in flowering and results in poor development of roots. Thus, decreased levels of miR393 are an indicator of stress tolerance in cotton and rice [137,138]. F-box proteins perform various activities during stress conditions such as degradation of proteins, rolling, and senescence of leaves [139,140]. It was found that miR394 prevents the translation of F-box genes family transcripts during abiotic stresses to maintain the optimal levels of proteins for normal functioning of plants, particularly in rice and *Arabidopsis* [141,142]. The miR395 regulates *APS* genes in response to various abiotic stresses. The major function of this gene family is the assimilation of sulphate [143].

## 4. Breeding Strategies for High Temperature Stress Tolerance

### 4.1. Conventional Breeding Approaches

Assessment of germplasm is a prerequisite for breeding stress tolerance. Numerous experiments have been conducted to identify heat stress tolerant genotypes from the available genepool. Moreover, the utilization of crop wild relatives is also gaining popularity in plant breeding due to their novel features that are lacking in domesticated cultivars. Most of these novel features are related to biotic and abiotic environmental stress. It is recommended to screen related wild species and relatives in order to have a diverse gene pool [144]. Although gene transfer from wild to cultivated species encounters numerous problems and is not always possible without recombinant DNA technology, the rapidly evolving technologies in plant sciences have made it quite possible to transfer genes among many species, as is discussed below [145]. After the identification of a suitable gene and trait, the next step is to transfer it to a desirable genotype or to purify the identified plant through selection. For this purpose, single plant selection, bulk selection, and pedigree selection are among the most widely used classical breeding methods in cotton [146,147]. These methods are used in cotton improvement along with molecular breeding tools for quick and efficient screening and genetic gain.

### 4.2. Molecular and Biotechnological Approaches

In addition to conventional screening and breeding approaches, molecular markers and biotechnological tools are also useful for improving stress tolerance of cotton genotypes [148]. Numerous markers such as amplified fragment length polymorphism (AFLP) and randomly amplified polymorphic DNA (RAPD) markers have been successfully utilized screening cotton genotypes for heat tolerance in the past [149,150]. Currently, simple sequence repeats (SSRs) and single nucleotide polymorphism (SNPs) are widely used markers for identifying quantitative traits loci (QTLs) related to stress tolerance in cotton [151,152]. The experiments were conducted by using heat tolerant and susceptible cultivars to determine heat responsive genes in upland cotton. Twenty-five expressed sequence tags (ESTs) were sequenced to study the homology of genes. The expression level of a few ESTs was also quantified using real time PCR. The results indicated that expression of folylpolyglutamate synthase (*FPGS$_3$*) and IAA-ala hydrolase (*IAR$_3$*) coding genes was significantly up-regulated during long-term and short-term high temperature stress. The expression of two non-annotated ESTs, i.e., *GhHS128* and *GhHS126*, was also found to be up-regulated under hot conditions. Thus, it was suggested that these two non-annotated ESTs are heat tolerant candidate genes [153].

In order to investigate the molecular mechanism of high temperature stress tolerance, the expression of some heat responsive genes was quantified through real time PCR in tolerant and susceptible upland cotton cultivars. The genes belong to various groups, i.e., *HSFA1b* and *HSFA2* are heat stress transcriptional factors; *HSP101*, *HSP70-1*, and *GHSP26* code for heat shock proteins; *ANNAT8* is involved in calcium signaling; and *APX1* controls antioxidant activity. The level of *GHSP26* increased in all genotypes, while the expression of *HSP101* and *HSP70-1* increased several-fold only in the seedlings of heat tolerant cultivars. The expression of *APX1* increased significantly in a heat-tolerant cultivar (VH-260), indicating the involvement of antioxidant activity in conferring heat tolerance. No significant change in the expression of *ANNAT8* was observed in heat susceptible cultivars. The expression of *HSFA2* and *HSFA1b* was several folds higher in leaves and ovaries of heat tolerant accessions than compared to heat susceptible accessions [154]. In order determine the SNP markers linked to the mitochondrial small heat shock protein (*MTsHSP*), a study was conducted by using accessions belonging to various cotton species, i.e., *G. aridum*, *G. sturtianum*, *G. gossypioides*, *G. stocksii*, *G. arboreum*, *G. laxum*, and *G. herbaceum*. Approximately 21 SNPs were identified for this gene by using PCR cloning and sequencing techniques, which could be useful for cotton improvement [155]. Transcriptomic analysis of 82 genes belonging to the *GhHSP20* family revealed their involvement in developmental and physiological processes of cotton. Most of them were regulatory in nature and expressed only under hot conditions, while eight genes were found to be involved in conferring tolerance for multiple stresses, namely heat, drought, and salinity [156].

Rapid advancements in applied genomics have resulted in useful tools for plant improvement. For example, markers linked to known genes or QTLs can be used for marker-assisted selection (MAS), as well as for genomic selection. Genomic selection assists the breeders in utilizing the molecular markers in the absence of phenotypic data. It can reduce the time for cultivar development through more efficient selection of progeny in early generations. An experiment was conducted with 550 recombinant inbred lines of cotton, and six fiber quality traits were evaluated using genomic selection. A total of 6292 markers were obtained through genotyping by sequencing. It was revealed that genomic selection could potentially predict genomic estimated breeding value in upland cotton fiber quality attributes [157]. Association mapping is also an effective technique for cotton improvement when information on population structure and linkage disequilibrium (LD) is available. This method is quite useful for reducing the laborious work involved in screening large populations [158]. Genome-wide association studies (GWAS) represent a powerful approach for identifying the locations of genetic factors that underlie complex traits. GWAS has been successfully implemented in cotton for the identification of single nucleotide polymorphism (SNP) loci and candidate genes for various attributes [159].

### 4.3. Transgenic Approaches

Transgenic techniques have also been extensively used to improve cotton cultivars for better tolerance with respect to high temperature stress. Recently, heat shock protein 70 (*AsHSP70*) from *Agave sisalana* was transformed in cotton through the *Agrobacterium* mediated transformation method. Expression studies showed the higher expression of transformed gene in different plant tissues under high temperature. Additionally, transgenic cotton plants exhibited improved performance for the measurable physiological and biochemical indicators [160]. In another study, over-expression of both *AVP1* and *OsSIZ1* genes in cotton improves lint yield compared to wild-type cotton under combined drought and heat stress conditions, and it does not have any negative affect on overall cotton yield when there is no stress. Furthermore, transgenic cotton plants also had 72% more photosynthetic rates two hours before the outset of heat stress and 108% higher photosynthetic rates during heat stress [161]. The function of *Arabidopsis* heat shock protein 101 (*AtHSP101*) is well known in heat tolerance at a vegetative stage, and the overexpression of this protein in cotton (Coker-312) clearly exhibited the increased germination percentage and enhanced pollen tube elongation under high temperatures than compared to non-transgenic cot-

ton [162]. Therefore, improved heat tolerance of reproductive systems in transgenic cotton is crucial for enhanced yield on a sustainable basis in the face of climate change. Conversely, the *Arabidopsis* SUMO E3 ligase (*AtSIZ1*) is a key gene for plant heat stress response, as *AtSIZ1* mutant plants exhibited increased susceptibility to high temperatures. In cotton, the overexpression of *OsSIZ1* gene, a rice homolog of *AtSIZ1*, conferred tolerance to both drought and heat stresses than compared with non-transgenic plants by demonstrating enhanced net photosynthesis rate and improved growth and development [163]. In another study, research revealed that the ectopic expression of Arabidopsis stress associated gene (*AtSAP5*) in transgenic cotton (Coker-312) protects several components of carbon gain and growth under extreme drought and associated heat stress conditions [164].

However, in order to render transgenic cotton more acceptable and efficient, it is necessary to focus on improving transformation efficiency. In a nutshell, these studies indicate that integrating heat stress-related genes in cotton is a viable strategy for engineering high-temperature tolerant cotton cultivars that could significantly improve lint yields in marginal environments, resulting in a sustainable cotton production. The effectiveness of heat stress responsive genes, for which its expression could be successful without the accompanying yield penalties, would determine the end degree of success.

### 4.4. CRISPR-Cas Mediated Genome Editing

Tolerance relative to high temperatures is mostly regulated by many genes based on the degree of stress and the stress tolerance mechanism. It will be more difficult to target a tolerance mechanism that is controlled by multiple genes. Plant heat stress response is precisely regulated by a complex web of TFs from distinct families. These TFs improve plant heat stress tolerance by modulating the expression of several stress responsive genes, either individually or in conjunction with other regulatory factors. There are numerous successful genetic engineering applications inducing heat stress tolerance in plants using heat stress TFs and HSPs genes [165,166]. However, genetically modified crop plants are subject to stringent regulatory requirements, which may cause lab research to be delayed in reaching the market. As an alternative to traditional transgenic approaches, recently emerged CRISPR-Cas-mediated genome editing allows researchers to alter, modify or swap alleles, and insert or silence gene(s) in a predefined manner [167].

High temperatures alter the expression pattern of several plant genes either by upregulating or down-regulating them. Although our understanding of differentially expressed genes in response to drought and salt stress has expanded, relatively less focus has been made on studying heat stress associated genes in cotton. Studying the expression pattern of heat stress responsive genes in cotton under long-term heat stress clearly showed that expressions of *HS126*, *HS128*, *FPGS*, *TH1*, and *IAR3* genes increased under high temperature. In contrast, the expressions of *ABCC3*, *CIPK*, *CTL2*, *LSm8*, and *RPS14* genes were downregulated [153]. Therefore, targeted modulation of these up-regulated and down-regulated genes in cotton using the CRISPR-Cas system would be an exciting opportunity for combating the negative impact of heat stress. Moreover, multiple HSPs and TFs associated with heat stress sensitive genes have been proposed as potential candidates for improving plant heat tolerance [168]. Therefore, understanding the exact role of these genetic regulators paves the way for the development of enhanced heat tolerance, while maintaining overall plant resilience. In maize plants, peak photosynthesis has been observed between 20 and 32 °C, and the subsequent increase in temperature caused a decrease in net photosynthesis rate depending on plant growth stage [169]. CRISPR/Cas9 mediated disruption of the heat-stress sensitive albino-1 (*HSA1*) gene in rice exhibits higher heat sensitivity compared to wild plants [170]. The *slagamous-like 6* (*Slagl6*) gene was reported as a potential gene in the study of facultative parthenocarpy. Thus, researchers have successfully developed heat-tolerant parthenocarpic tomato fruit by mutating the *SlAGL6* gene using the state-of-the-art CRISPR-Cas9 system [171]. In addition, thermosensitive male sterile maize lines have also been developed by mutating the *thermosensitive genic male-sterile 5* (*TMS5*) gene using CRISPR-Cas9 editing [172].

CRISPR-Cas9 has been modified and exploited for a range of new functions, including controlling gene regulation by activating and suppressing target gene expression using CRISPR activation and interference systems [173]. Positive gene regulators associated with HSPs and stress related TFs could be activated through the CRISPR activation system with high specificity. Moreover, negative regulators could be knocked out by using the CRISPR interference system. In one of study, the *BZR1* gene was up-regulated and repressed using CRISPR activator and interference systems. The results show that the overexpression of the *BZR1* gene enhances $H_2O_2$ production and recovery of thermo tolerance in rice, while plants with suppression of the gene show impaired production of $H_2O_2$ in apoplast and reduced heat tolerance [174]. Previously, the roles of *MAP3Ks* remained poorly understood in cotton. Recently, it has been reported that *MAP3K65* gene expression is induced by multiple signaling molecules, pathogen infection, and heat stress. This gene enhances susceptibility to pathogen infection and heat stress by negatively modulating growth and development related processes. Moreover, silencing *GhMAP3K65* enhanced resistance to pathogen infection and heat stress in cotton. Therefore, *GhMAP3K65* is a potential candidate gene to target with the CRISPR-Cas9 genome editing system in order to engineer heat tolerance in cotton [175]. In conclusion, maneuvering positive and negative regulators of heat stress signaling molecules in cotton could, thus, be exploited to develop new cotton cultivars tolerant to extreme temperatures.

## 5. Conclusions and Future Directions

High temperature exerts a negative impact on cotton growth and yield in all stages of plant growth. It reduces lint quantity and quality by impeding normal plant biological processes and pathways. Understanding the heat tolerance mechanism and molecular characterization of related genes is essential for developing stress tolerant cultivars for sustainable cotton production under changing climatic conditions. Undoubtedly, the heat stress TFs, HSPs, and other genes are important for maintaining the secondary structure of proteins, and rapid sensing of heat is also critical to induce the protective mechanisms against high temperature stress. Traditional breeding approaches for developing stress tolerance are being complemented by new technologies, including state-of-the-art genome editing tools, speed breeding approaches, and various omics tools, which may decrease the time needed to develop cotton cultivars with increased heat tolerance. For engineering high temperature stress tolerance, the CRISPR-Cas system is considered a non-genetically modified (nGM) approach, thus allowing for the expansion of scientific community efforts to introduce heat stress tolerance in future cotton cultivars for all cotton growing regions. Furthermore, the increasing demand of high-quality lint yield in a rapidly growing world population will be confronted through concept of speed breeding. It will aid quick generation advancement in order to shorten the overall growth cycle and accelerate cotton breeding programs. In addition, recent advances in sequencing technologies have made significant strides in successfully sequencing complex cotton genome. Subsequently, applications of various omics approaches have pointedly enhanced our understanding of cotton physiology and the functions of genes in response to heat stress. Therefore, differently expressed genes, proteins, and metabolites identified through different omics tools can be used as a potential biomarker to develop high temperature-resilient cotton cultivars.

**Author Contributions:** Conceptualization, M.T.A., S.M., L.H., Y.J. and M.S.M.; writing and draft preparation, S.M., M.T.A. and M.S.M. writing, review and editing, M.T.A., R.M.A., S.-H.Y. and G.C.; supervision, X.D., I.A.R. and M.T.A. All authors have read and agreed to the published version of the manuscript.

**Funding:** This research received no external funding.

**Institutional Review Board Statement:** Not applicable.

**Informed Consent Statement:** Not applicable.

**Acknowledgments:** The presented study is part of research proposal "Genotyping and development of heat stress tolerant cot-ton germplasm having enhanced quality traits" No. 964, and the authors are grateful to the Center for Advanced Studies and Punjab Agricultural Research Board (CAS-PARB), Pakistan, for providing funds for this study.

**Conflicts of Interest:** The authors declare no conflict of interest.

**Disclaimer:** USDA is an equal opportunity provider and employer.

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
