# Peer review of "Heat Stress in Cotton: A Review on Predicted and Unpredicted Growth-Yield Anomalies and Mitigating Breeding Strategies"

_agronomy, doi:10.3390/agronomy11091825_

Round 1
Reviewer 1 Report
The review is interesting to publish. According to me, you have to add some more data for heat effect on yield and quality of cotton fibre (paragraph 2.4).
Also you have to take care of the below issues
- 1.1 title is not needed. The paragraph can be part of introduction
- line 69 add cotton befor "plant"
- reference 26, add information for the paper or book
- line 210 - explain ROS
- line 221- is Figure 3 and not 2.3
- line 293, Table 1 it will be better to be one page
- in the references session take care not to use capital letters in some titles
Author Response
Hi,
Response to reviewer 1 is attached herewith.
Authors are thankful to editors and reviewers for their valuable comments and suggestions which have been added in the manuscript in “Red color text and track changes ”.

Reviewer 2 Report
I find the review to be rather superficial and the content does not reflect the title at all. In no way can this review be considered interactive, and there is little to nothing on growth-yield anomalies or mitigating breeding strategies.
Since this review is specifically on cotton it is essential for the authors to indicate what is known in general in plants with regards to heat stress and what is unique to cotton or currently unknown in cotton. This review does not distinguish between these things and the review is full of non-cotton publications which you wouldn’t know from the text of the review unless you look at the papers cited. Therefore, if another species is cited, is this because this information is not known in cotton, does this represent an information gap, which if true would be an important contribution for this review.
The title of the review also just says cotton, which is problematic as some species of cotton are rather well known for their heat tolerance eg G. arboreum, whereas other species such as G hirsutum are not. There is little to no mention of this in the review, and where cotton is mentioned it should be indicated what species the work cited relates to as it makes a significance difference.
Cotton species are generally perennial and hot climate species, which makes it quite different from the majority of the literature that is derived from annual and cooler climate species like Arabidopsis. Therefore, strategy mechanisms and tolerance of cotton to heat stress are likely to be potentially quite different to other species, hence the review should be grounded mostly in cotton related research. Many of the references I find baffling why that work of that species was chosen.
The approaches for improving cotton heat tolerance is one of the weakest areas of the review with no idea whether anything has really actually improved performance. There has been a large array of transgenic experiments with a goal to improve tolerances of many areas including heat stress that are not mentioned at all in this review.
There is also no mention of genomic prediction/selection which is probably one of the few markers related methods that has the possibility of coping with the complexity of this trait and also its significant envirnomental aspects.
There are many areas of the review that are poorly written and use very colloquial English with several typos present eg line 376 god instead of good.
Overall, I find this review quite poor, not very helpful to the reader and certainly not related in any way to its title. A significant revision.
Author Response
Response to reviewer-2 is attached herewith

Round 2
Reviewer 2 Report
The review has been significantly improved, especially the English.
The title of this review is still quite misleading as there is nothing that is interactive about it.
My major problem with this review is that it still references work from other species, but gives the impression in the text that that reference is associated with cotton. For instance in the section entitled 2. Effects of High Temperature Stress on cotton ,lines 79 to 81 states ‘Both day and night temperatures play an important role in determining yield potential but high night temperature reduces yield and causes significant damage while the role of high day temperature is secondary (16)’. Reference 16 refers to Willits D, Peet M. The effect of night temperature on greenhouse grown tomato yields in warm climates. J 553 Agricultural Forest meteorology. 1998;92(3):191-202.
I think therefore that this is misleading, and the reference has nothing at all to do with cotton. If the authors would still like to reference this work, they should state ‘that in tomato it was found etc’. This type of referencing of other species when the text appears to indicate its related to cotton should be changed so that the text indicates the species that the work references when it is not from cotton.
There have been many transgenic papers in cotton using genes from other species that appear to improve cottons heat stress but none of this literature is mentioned in this review.
Author Response
The response to reviewer is attached

Round 3
Reviewer 2 Report
The authors have sufficiently addressed my critisims of the review. There remains some typos that require attention.
Author Response
Hi,
The response to editor/reviewer is attached herewith
Regards,
Dr. Muhammad Tehseen Azhar
